

# Molecular Dynamics Simulation of the Surface Tension of Aqueous Sodium Chloride: from Dilute to Highly Supersaturated Solutions and Molten Salt

Xiaoxiang Wang[1], Chuchu Chen[1], Kurt Binder[2], Uwe Kuhn[1], Ulrich Pöschl[1], Hang Su[3,1]*, Yafang Cheng[1,3]*

[1] Max Planck Institute for Chemistry, Multiphase Chemistry Department, Hahn-Meitner-Weg 1, 55128 Mainz, Germany

[2] Institut für Physik, Johannes Gutenberg-Universität, Staudinger Weg 7, 55128 Mainz, Germany

[3] Institute for Environmental and Climate Research, Jinan University, 510632 Guangzhou, China

* Correspondence to: Yafang Cheng (yafang.cheng@mpic.de) and Hang Su (h.su@mpic.de)

**Abstract.** Sodium chloride (NaCl) is one of the key components of atmospheric aerosols. The surface tension of aqueous NaCl solution ($\sigma_{NaCl,sol}$) and its concentration dependence are essential to determine the equilibrium water vapor pressure of aqueous NaCl droplets. Supersaturated NaCl solution droplets are observed in laboratory experiments and under atmospheric conditions, but the experimental data for $\sigma_{NaCl,sol}$ are mostly limited up to sub-saturated solutions. In this study, the surface tension of aqueous NaCl is investigated by molecular dynamics (MD) simulations and pressure tensor method from dilute to highly supersaturated solutions. We show that the linear approximation of concentration dependence of $\sigma_{NaCl,sol}$ at molality scale can be extended to the supersaturated NaCl solution until a molality of ~9.6 mol kg$^{-1}$ (i.e., solute mass fraction ($x_{NaCl}$) of ~0.36). Energetic analyses show that this monotonic increase of surface tension is driven by the increase of excessive surface enthalpy ($\Delta H$) as the solution becomes concentrated. After that, the simulated $\sigma_{NaCl,sol}$ remains almost unchanged until $x_{NaCl}$ of ~0.47 (near the concentration upon efflorescence). The existence of the "inflection point" at $x_{NaCl}$ of ~0.36 and the stable surface tension of $x_{NaCl}$ between ~0.36 and ~0.47 can be attributed to a competitive growth of excessive surface entropy term ($T \cdot \Delta S$) and the excessive surface enthalpy term ($\Delta H$). After a "second inflection point" at $x_{NaCl}$ of ~0.47, the simulated $\sigma_{NaCl,sol}$ gradually regains the growing momentum with a tendency to approach the surface tension of molten NaCl (~148.4 mN m$^{-1}$ at 298.15 K, MD simulation based extrapolation). This fast increase of $\sigma_{NaCl,sol}$ at $x_{NaCl}$ > 0.47 is primarily still an excessive surface enthalpy-driving process, although contribution from concurrent fluctuation of excessive surface entropy is expected but in a relatively smaller scale. Our results reveal different regimes of concentration dependence of the surface tension of aqueous NaCl at 298.15 K: a water-dominated regime ($x_{NaCl}$ from 0 to ~0.36), a transition regime ($x_{NaCl}$ from ~0.36 to ~0.47) and a molten NaCl-dominated regime ($x_{NaCl}$ from ~0.47 to 1).

## 1. Introduction

Sodium chloride (NaCl) is one of the most important components of atmospheric aerosol particles (Finlayson-Pitts, 2003; Lewis and Schwartz, 2004). The aqueous NaCl solution droplet could



participate in many atmospheric processes, such as phase transition, cloud activation, ice crystallization, long-range transport and chemical aging (Martin, 2000; Finlayson-Pitts, 2003; Ghorai et al., 2014; Wagner et al., 2015; Chen et al., 2016). To better understand these processes, the concentration-dependent surface tension of aqueous NaCl solution ($\sigma_{NaCl,sol}$) is essential to determine

the equilibrium between NaCl solution droplet and water vapor (Jarvis and Scheiman, 1968; Dutcher et al., 2010).

Below saturation point (~6.15 mol kg$^{-1}$), $\sigma_{NaCl,sol}$ shows a near linear dependence on molality (Jarvis and Scheiman, 1968; Johansson and Eriksson, 1974; Aveyard and Saleem, 1976; Weissenborn and Pugh, 1995; Matubayasi et al., 2001) with a slope of 1.73±0.17 (Pegram and Record, 2006, 2007).

Because of the energy barrier of crystallization during dehydration and size-effects at nanoscale (Martin, 2000; Biskos et al., 2006; Cheng et al., 2015), supersaturated aqueous NaCl solution droplets can exist under atmospheric conditions. However, direct measurements of surface tension of supersaturated droplets are challenging due to technical difficulties (Harkins and Brown, 1919; Vargaftik et al., 1983; Richardson and Snyder, 1994; Kumar, 2001). Only recently, Bzdek et al. (2016)

overcame this limitation with an optical tweezer method and extend the concentration range to ~8 mol kg$^{-1}$, where the near linear relationship still holds (Bzdek et al., 2016).

It is a matter of debate to which extent the approximation of a near linear dependence of surface tensions on molality can still be used for NaCl droplets. Cheng et al. (2015) used the Differential Köhler Analyses (DKA) method to retrieve the surface tension of NaCl aqueous droplets, and revealed

a large deviation from the near linear increase at molality of ~10 mol kg$^{-1}$. In literature, such deviation in concentrated solution has also been found for other compounds, such as HNO$_3$ (Weissenborn and Pugh, 1996) and it is believed to be typically true for most highly soluble electrolytes (Dutcher et al., 2010). The reason for such deviation remains unclear.

A few surface tension models have been developed for highly concentrated electrolyte solutions,

e.g., Li and Lu (2001), Li et al. (1999), Levin and Flores-Mena (2001). Li and Lu (2001) developed a model based on the Gibbs dividing surface concept, where the adsorption and desorption rate constants, saturated surface excess, stoichiometric coefficient of ions and mean ionic activity coefficient are needed. For NaCl aqueous solution, this model is suitable for solution with concentration up to ~5.5 mol kg$^{-1}$. Li et al. (1999) uses Debye-Huckel parameter, osmotic coefficient and a proportionality

constant from the fitting of measured values to calculate the surface tension, which covers the concentration until saturation point of bulk NaCl aqueous solutions. The remaining models are mostly only suitable for the dilute electrolyte solutions, such as the one proposed by Levin and Flores-Mena (2001). In their valid concentration range, these surface tension models produce linear or near linear concentration dependence of $\sigma_{NaCl,sol}$ that agrees well with currently available observations.

One surface tension model that is able to predict $\sigma_{NaCl,sol}$ in the whole concentration range from infinitely dilute ($x_{NaCl} = 0$) to highly supersaturated solution to molten salts ($x_{NaCl} = 1$) was proposed by Dutcher et al. (2010), which has been adopted into the widely used Extended Aerosol Inorganics Model (E-AIM) (Wexler and Clegg, 2002). This model is based on the concept: ions are solvated by the water at low salt concentrations, meaning that water-structures like hydration shells are formed

around the ions; while at very high salt concentration the water is considered as "solute" hat is solvated




by the ions instead, meaning a salt-structure is formed around the water molecules. Accordingly, for a diluted solution, the surface tension of water dominates and the surface tension of the solution equals the surface tension of water adjusted by a term that is proportional to the solute concentration. For a highly supersaturated solution, a similar relationship can be applied with the surface tension of molten

salt as governing element. Legitimately, the model is then constrained by the surface tensions of water and molten salt. The parameterization of this model is obtained by fitting the data of sub-saturated solutions. When the aqueous NaCl solution gets concentrated, this model shows a nonlinear monotonically increasing trend of $\sigma_{NaCl,sol}$ generally in good agreement with observations, but no "inflection point" was introduced. It should be noted that the surface tension as a function of mole

fraction of NaCl according to the Dutcher et al. (2010) model is essentially a linear interpolation between the surface tensions of water and molten NaCl.

In this study, we applied molecular dynamics (MD) simulations and pressure tensor method to calculate the concentration dependence of $\sigma_{NaCl,sol}$ from infinitely dilute ($x_{NaCl} = 0$) to highly supersaturated solution to molten salt ($x_{NaCl} = 1$). The concentration dependence of $\sigma_{NaCl,sol}$ is

divided into 3 regimes: a water-dominated regime, a transition regime and a molten NaCl-dominated regime. We compare our results with the Dutcher et al. (2010) model, and present the principal underlying physical chemistry (driving forces) behind the change of surface tension along concentration changes.

## 2.  Methods

### 2.1 MD simulation

MD simulations were carried out with the GROMACS 5.1 package (Abraham et al., 2015). The $Na^+$ ions, $Cl^-$ ions and water molecules were added into a cubic box ($L = 5$ nm) to imitate the NaCl solution. The concentrations of simulated solutions are summarized in Table 1. To simulate the surface

tension of supersaturated NaCl aqueous solution, we make use of the time window in the MD simulations before the crystallization starts in the system. The highest $x_{NaCl}$ we can reach is ~0.64 (the corresponding concentration is ~30.39 mol kg$^{-1}$), below which the simulated surface tensions in three independent runs stably converge after 50 to 100 ns (Fig. 1). For more concentrated solutions, stable convergence cannot be reached, as for example large fluctuations are shown in Fig. 1d at $x_{NaCl}$ of 0.75.

According to Dutcher et al. (2010), surface tension of liquid/molten NaCl at 298.15 K (corresponding $x_{NaCl}$ is 1, infinite concentrated solution) can be regarded as the upper boundary of $\sigma_{NaCl,sol}$. We simulated surface tension of molten NaCl at high temperature (i.e., from 1000 K to 1700 K) and extrapolated them back to 295.15 K, as it has been done to determine the surface tension of molten salts at room temperature based on experimental observations (Dutcher et al., 2010).

The procedure (Fig. 2) of simulation we followed is: (1) systems were firstly energetically minimized by the steepest-descent method (Stillinger and Weber, 1985) (2) Solutions were equilibrated in the *NVT* ensemble and *NPT* ensemble (pressure = 1 bar) with periodic boundary conditions in three directions. The temperature was controlled by using the velocity-rescaling thermostat (Bussi et al., 2007). The box volume change due to the variation of density at different temperatures, and in our case

the length of cubic box varied from 4.9 nm to 5.1 nm. (3) The box was elongated along the z-direction





with $L_z$ = 20 nm to create two interfacial regions. (4) The solution was equilibrated and simulated with the *NVT* ensemble in the rectangular parallelepiped box at the corresponding temperature. (5) Systems without surfaces were also simulated for further energy analysis, and the trajectories obtained from step 2 were simulated with *NPT* ensemble. (6) All simulations were carried out for at least 200 ns, which is much longer than that in previous studies (a few nanoseconds, Jungwirth and Tobias, 2000; Neyt et al., 2013) because the system that we were dealing with is much more concentrated. 1 fs time step was adopted and conformations for analysis were saved every 2 ps. Both electrostatic interactions and van der Waals interactions were calculated using the particle mesh Ewald (PME) algorithm, which has been proven to be a good choice for accurate calculation of long-range interactions (Essmann et al., 1995; Fischer et al., 2015). To test the reproducibility, all the systems were simulated 3 times, and the respective statistical error bars were provided.

In our simulation, the Joung-Cheatham (JC) force field for NaCl (Joung and Cheatham III, 2009) with SPC/E water model (Berendsen et al., 1987) was applied to simulate the NaCl solution and molten NaCl. The solubility at 298.15 K based on JC force field with SPC/E model has been determined as 3.7±0.2 mol kg$^{-1}$ (Paluch et al., 2010; Aragones et al., 2012 ; Espinosa et al., 2016), which to our best knowledge is the most close one to the experimental value of solubility (~6.15 mol kg$^{-1}$)

### 2.2 Calculation of Surface Tension

Based on results from MD simulations, the surface tension was calculated by using the mechanical definition of the atomic pressure (Alejandre et al., 1995):

$$\sigma_{MD} = 0.5L_z[\langle P_{zz}\rangle - 0.5(\langle P_{xx}\rangle + \langle P_{yy}\rangle)] \tag{1}$$

where $\sigma_{MD}$ can represent the surface tension of molten NaCl ($\sigma_{NaCl}$), NaCl solution ($\sigma_{NaCl,sol}$) or pure water ($\sigma_{water}$), $L_z$ is the length of the simulation cell in the longest direction (along z-axis) and $P_{aa}$ (a=x, y, z) denotes the diagonal component of the pressure tensor. The $\langle ...\rangle$ refers to the time average. The factor 0.5 outside the squared brackets takes into account the two interfaces in the system. Only the stable values were taken as our calculated surface tension.

### 2.3 Energy analysis

The excessive surface enthalpy denotes the additional enthalpy in the system due to the creation of surfaces. It can be calculated as the difference of enthalpy between solutions with and without surfaces (Bahadur et al., 2007),

$$\Delta H = H_{b\_s} - H_b \tag{2}$$

where $H_{b\_s}$ is the total enthalpy of simulated systems with surfaces and $H_b$ is the total enthalpy of simulated systems without surfaces. As the kinetic energy is the same for systems with or without surfaces and the difference of $pV$ can be ignored, $\Delta H$ can be presented as

$$\Delta H = E_{b\_s} - E_b \tag{3}$$

where $E_{b\_s}$ and $E_b$ are the potential energy of the system with and without surfaces.

Then the surface tension can be determined by the excessive surface free energy per unit area as in Eq. (4) (Davidchack and Laird, 2003):





$$\sigma = \frac{\Delta G}{A} = \frac{\Delta H - T \cdot \Delta S}{A} \qquad (4)$$

where $\Delta G$ is the increased part of free energy due to the creation of surfaces, $A$ is the total area of the surface we created, so $A = 2 \times a$ and $a$ is the area of each created surface. $\Delta S$ is the excessive surface entropy. We then can retrieve $\Delta S$ by using the data of enthalpy and surface tension:

$$\Delta S = \frac{\Delta H - \sigma \cdot A}{T} \qquad (5)$$

$\Delta H$ and $T \cdot \Delta S$ per unit area ($\frac{\Delta H}{A}$ and $\frac{T \cdot \Delta S}{A}$) are obtained as the enthalpic and entropic part of contributions to the net surface tension, which will be used to explain the change of surface tension along with the mass fraction of NaCl ($x_{NaCl}$).

### 3. Results and Discussion

#### 3.1 Water-dominated regime ($x_{NaCl} < \sim 0.36$)

In Fig. 3a, the calculated surface tension of NaCl aqueous solution ($\sigma_{NaCl,sol}$) are compared with experimentally determined values (Jarvis and Scheiman, 1968; Johansson and Eriksson, 1974; Aveyard and Saleem, 1976; Weissenborn and Pugh, 1995; Matubayasi et al., 2001; Pegram and Record, 2006,

2007; Morris et al., 2015; Bzdek et al., 2016) in the sub-saturated concentration range (molality of NaCl solution from 0 to 6.15 mol kg[-1] and $x_{NaCl}$ from 0 to ~0.265). At 298.15 K, both model simulation (red solid points and fit line in Fig. 3a) and experimental observation (black line in Fig. 3a) reveal a linear dependence of surface tension on solution concentration at molality scale, with a very similar slope (1.9 versus 1.73±0.17, respectively). Systematic underestimation, however, exists in the

simulated $\sigma_{NaCl,sol}$. The previous MD simulations by Neyt et al. (2013) has also reported a similar result for the solution whose concentration ranges from 0 to 5.2 mol kg[-1] by using the same water model (SPC/E) but two different NaCl force fields, i.e., Wheeler NaCl (solid dark blue triangle in Fig. 3a) and Relf NaCl (open light blue triangle in Fig. 3a). Bhatt et al. (2004) also used the Wheeler NaCl model and SPC/E water model revealing a linear dependence and underestimation. When we subtract

the experimentally determined and the MD simulated surface tension of pure water ($\sigma_{water}$) from the observed and modeled $\sigma_{NaCl,sol}$, respectively, the relative increase of surface tension ($\Delta\sigma = \sigma_{NaCl,sol} - \sigma_{water}$) from models and experiments converge nicely (Fig. 3b). The MD simulation is able to reproduce the increment in the growth of surface tension from pure water due to the addition of solute NaCl though the predicted absolute value of $\sigma_{NaCl,sol}$ is systematically underestimated, which

may mainly be attributed to the discrepancy between observed $\sigma_{water}$ and the modeled ones from the SPE/C water model.

By performing MD simulations in the supersaturated concentration range, we found that this linear relationship still holds beyond the saturation point until $x_{NaCl}$ of ~ 0.36 (Fig. 4). As mentioned above, the laboratory experiments with elevated NaCl aqueous droplet and the optical tweezer method show

that the linear relationship between $\sigma_{NaCl,sol}$ and NaCl concentration (molality scale) can be extended to ~ 8 mol kg[-1] (Fig. 3) (Bzdek et al., 2016), corresponding to $x_{NaCl}$ of ~ 0.33 (Fig. 4), which is consistent with our simulations.





### 3.2 Transition regime ($x_{NaCl}$ from ~0.36 to ~0.47)

Surface tensions of single inorganic electrolyte aqueous solution were often found to be linear functions of concentration (at the molality scale) over moderate concentration range (Talbot, 1987). However, these simple relationships will not hold when the solutions become more concentrated. As shown in Fig. 4, starting from $x_{NaCl}$ ~0.36, the simulated $\sigma_{NaCl,sol}$ remains almost unchanged until $x_{NaCl}$ of ~0.47 (concentration upon efflorescence). This "inflection point" of $\sigma_{NaCl,sol}$ at $x_{NaCl}$ of ~0.36 is supported by those determined by the DKA approach (Cheng et al., 2015), where a large deviation of surface tension from the monotonic linear increase. Note that beyond $x_{NaCl}$ of ~0.47, the simulated surface tension increases again (Fig. 4). This "second inflection point", right at the concentration upon efflorescence, may imply potential correlation with crystallization processes.

### 3.3 Molten NaCl-dominated regime ($x_{NaCl} > ~0.47$)

Beyond the "second inflection point" ($x_{NaCl} > 0.47$), the simulated $\sigma_{NaCl,sol}$ gradually regains a growing momentum (Fig. 4). Unfortunately, due to the large fluctuation in the surface tension simulation (Fig. 1), we are not able to extend our surface tension calculation in this way beyond $x_{NaCl}$ of ~0.64. However, according to Dutcher et al. (2010), it is expected that the surface tension of the solution would ultimately approach the surface tension of the hypothetical molten solute (i.e., $x_{NaCl} = 1$) at the same temperature. This hypothesis has been found to be consistent with the DKA retrieval for a highly concentrated ammonium sulfate aqueous solution with molality of ~380 mol kg[-1] (Cheng et al., 2015). We thus also try to constrain the growth of $\sigma_{NaCl,sol}$ by MD simulated surface tension of molten NaCl ($\sigma_{NaCl}$) at 298.15 K.

Similar to Janz (1988)'s experimental results, the simulated $\sigma_{NaCl}$ is also linearly correlated with temperature from 1000 K (the simulated melting point of NaCl) to 1700 K, as shown in Fig. 5a. Following Dutcher et al. (2010), a surface tension of ~148.4 mN m[-1] is obtained for the hypothetical molten NaCl at 298.15 K by linear extrapolation of the MD simulated $\sigma_{NaCl}$ at higher temperature, which is very close to the ~169.7 mN m[-1] extrapolated from the experimental results (Dutcher et al., 2010). Combined with $\sigma_{NaCl} = \sigma_{NaCl,sol}(x_{NaCl} = 1) = ~148.4$ mN m[-1], the simulated $\sigma_{NaCl,sol}$ in the concentration range of $x_{NaCl} > 0.47$ shows the tendency to ultimately approaching the surface tension of melting NaCl at 298.15 K, similar to the blue curve in Fig. 4 from the Dutcher et al. (2000) study.

### 3.4 Physical chemistry behind the regimes

In energetic analyses, surface tension was decomposed into excessive surface enthalpy ($\frac{\Delta H}{A}$) and excessive surface entropy ($\frac{T \cdot \Delta S}{A}$). Note that the increase in excessive surface entropy ($\frac{T \cdot \Delta S}{A}$) or decrease of $-\frac{T \cdot \Delta S}{A}$, respectively, will contribute negatively to the growth of $\sigma_{NaCl,sol}$. The analyses show that the monotonic increase of surface tension in water-dominated concentration range ($x_{NaCl}$ from 0 to ~0.36) is driven by the increase of $\frac{\Delta H}{A}$ when the solution becomes concentrated (Fig. 6). When the solution gets concentrated, $\frac{\Delta H}{A}$ first increases slightly with enhanced increasing rate when approaching



saturation (at $x_{NaCl}$ >~0.2) and in the supersaturated regime up to $x_{NaCl}$ of ~ 0.36. $-\frac{T \cdot \Delta S}{A}$ behaves differently, it remains almost constant at about -45 mN m$^{-1}$ first and only starts to decrease at $x_{NaCl}$ ~ 0.2. This way, in this concentration range ($x_{NaCl}$ from 0 to ~ 0.36), the increase of excessive surface enthalpy outnumbers the increase of excess surface entropy and thus this physicochemical regime can

be understood as an excessive surface enthalpy-driving process.

The stable surface tension in the transition-regime concentration range ($x_{NaCl}$ from ~0.36 to ~0.47) is attributed to a competitive growth of excessive surface entropy term ($-\frac{T \cdot \Delta S}{A}$) along the retarded increase in $\frac{\Delta H}{A}$. Figure 6 shows that in the concentration below $x_{NaCl}$ of ~0.36, the increase of $\frac{\Delta H}{A}$ significantly slows down and stabilizes at ~145 mN m$^{-1}$ when the mass fraction approaches the

efflorescence point. During this period, $-\frac{T \cdot \Delta S}{A}$ keeps decreasing and the changes is almost comparable to that that of $\frac{\Delta H}{A}$, which results in a corresponding $\sigma_{NaCl,sol}$ almost independent to the solution concentration change.

Here, we present a potential explanation for the stability of surface tension in this region from the structural analysis. The ratio of Na$^+$ concentration at different positions to the average concentration of

the whole system ($C_z/C_{average}$) in different solutions is shown in Fig. 7a. The three blue-toned lines represent the ratio of solution in the transition regime with $x_{NaCl}$ from ~0.36 to ~0.47. All of them have apophysises (significant rise) near the surface and these apophysises almost overlap with each other. This phenomenon suggests that the solute in these solutions enriches close to the surface and the degree of enrichment is almost the same for the different-concentration solution. Here, we denote the

significant difference of the solute concentration in bulk region and on surface as a type of liquid-liquid partitioning. To check if this partitioning is dependent on the size of solution slab, we calculate the corresponding value of a 3 nm × 3 nm × 10 nm solution slab with $x_{NaCl}$ of 0.4 (Fig. 7b). There is still an apophysis near surface, thus we can claim that the partitioning is independent of the size of solution slab in the simulation. Note that this surface enrichment of NaCl does not mean that NaCl is enriched

right on top of the solution surface. Actually the density profile of water extends about 0.2 nm beyond that of NaCl towards the vapor region. By contrast, the solution with $x_{NaCl}$ >0.47 or <0.36 do not have this type of partitioning as shown by the red and green lines. This comparison implies that the stability of surface tension of solution with $x_{NaCl}$ from ~0.36 to ~0.47 is related to the "bulk-surface" partitioning. This interpretation is only a conjecture, and more studies are needed to further examine

this phenomenon and interpretation. The shallow minimum in the density profile for $x_{NaCl}$ between 0.36 and 0.47 to the left of the maximum is somewhat unexpected, and one might expect equilibration problems. However, we have checked that this structural feature develops already during the first 10 ns of the MD simulation, and does not change at all during the residual 200 ns. Surface enrichment of NaCl can be expected, however, when the solubility limit of the water-rich solution in the bulk is

reached. Very roughly, such phenomena are analogous to interfacial wetting phenomena such as surface melting of crystals (Frenken and Van der Veen, 1985), which sometimes is observed when the temperature is raised towards the triple point. In our case, the enrichment zone of NaCl (which is about



0.4 nm thick in Fig.7) would be a precursor effect to the (metastable) NaCl-rich bulk solution.

As shown in Fig. 6, when a solution gets more concentrated from $x_{NaCl}$ of ~0.47 to ~0.64, the $\frac{\Delta H}{A}$ slightly increases from the plateau of ~145 mN m$^{-1}$ but the change is only ~5 mN m$^{-1}$. In contrast, the $-\frac{T \cdot \Delta S}{A}$ starts to increase ~5 mN m$^{-1}$ (~-62 mN m$^{-1}$ to ~-57 mN m$^{-1}$). So during this period, both surface

excessive enthalpy term and entropy term contribute to the growth of $\sigma_{NaCl,sol}$. To constrain the energetic analyses, the $\frac{\Delta H}{A}$ and $\frac{T \cdot \Delta S}{A}$ were also calculated for the molten NaCl at 298.15 K by linearly extrapolate the corresponding values of $\frac{\Delta H}{A}$ and $\frac{T \cdot \Delta S}{A}$ simulated at higher temperature (1000-1700 K, Fig. 5b). By combing $\frac{\Delta H}{A}(x_{NaCl} = 1)$ = 206.1 mN m$^{-1}$ and $\frac{T \cdot \Delta S}{A}(x_{NaCl} = 1)$ = 57.7 mN m$^{-1}$ into Fig. 6, it is expected that excessive surface enthalpy term will still have a large space (about more than 55 mN m$^{-1}$)

to grow until approaching $\frac{\Delta H}{A}$ of molten NaCl at 298.15 K. But the surface excessive entropy of molten NaCl at 298.15 K is similar to that at $x_{NaCl}$ of ~0.64. Thus, the fast increase in $\sigma_{NaCl,sol}$ in the concentration of $x_{NaCl}$ from ~0.47 to 1 can be assumed to be primarily an excessive surface enthalpy-driving process, although a contribution of excessive surface entropy is expected but in a relatively smaller scale.

### 4.   Conclusion

The analysis based on the calculated surface tension confirms the basic concept of the Dutcher et al. (2010) semi-empirical model, while unfold a more detailed global landscape of concentration dependence of surface tension of aqueous NaCl solution and its driving forces: (1) a water-dominated

regime ($x_{NaCl}$ from 0 to ~0.36, at low concentrations the ion is solvated by the water, meaning that water-structures/hydration shells are formed around the ion); (2) a transition regime ($x_{NaCl}$ from ~0.36 to ~0.47); and (3) a molten NaCl-dominated regime ($x_{NaCl}$ from ~0.47 to 1, at very high salt concentration water is solvated by the ions meaning that a salt-structure is formed around the water molecules). Note that our result may not exactly reflect the real mode of surface tension of NaCl

solution along the concentration, but it does imply the concept of a non-monotonic change of surface tension.

### 5.   Data availability

Readers who are interested in the data should contact the authors: Yafang Cheng

(yafang.cheng@mpic.de), Hang Su (h.su@mpic.de) or Xiaoxiang Wang (xiaoxiang.wang@mpic.de).

### Acknowledgement

This study is supported by the Max Planck Society (MPG). Xiaoxiang Wang acknowledges the support from China Scholarship Council (CSC, 201406190170). Yafang Cheng acknowledges the Minerva

Program from MPG.



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



Table 1. Concentrations of solution studied in our simulation and the calculated values of surface tension.

| NO. | Number of water | Number of NaCl | Concentration (mol kg$^{-1}$) in bulk region [b] | $x_{NaCl}$ in bulk region | Concentration (mol kg$^{-1}$) of whole solution | $x_{NaCl}$ of whole solution | Surface tension (mN m$^{-1}$) |
|---|---|---|---|---|---|---|---|
| 1 | 4142 | 0 | 0 | 0 | 0 | 0 | 61.9±0.02 |
| 2 | 4058 | 42 | 0.657 | 0.037 | 0.575 | 0.0325 | 63±0.24 |
| 3 | 3976 | 83 | 1.235 | 0.067 | 1.159 | 0.0635 | 63.9±0.14 |
| 4 | 3824 | 159 | 2.41 | 0.123 | 2.309 | 0.119 | 66.23±0.1 |
| 5 | 3728 | 207 | 3.16 | 0.156 | 3.08 | 0.1528 | 67.56±0.17 |
| 6 | 3656 | 243 | 3.85 | 0.184 | 3.69 | 0.1776 | 68.93±0.06 |
| 7 | 3550 | 296 | 4.8 | 0.219 | 4.63 | 0.213 | 70.67±0.1 |
| 8 | 3452 | 345 | 6.04 | 0.261 | 5.552 | 0.245 | 73±0.087 |
| 9 | 3388 | 377 | 6.75 | 0.283 | 6.182 | 0.265 | 73.93±0.37 |
| 10 | 3314 | 414 | 7.47 | 0.304 | 6.94 | 0.288 | 75.8±0.25 |
| 11 | 3222 | 460 | 8.57 | 0.334 | 7.931 | 0.317 | 78.13±0.73 |
| 12 | 3108 | 517 | 9.745 | 0.36 | 9.24 | 0.351 | 79.58±0.38 |
| 13 | 3038 | 552 | 10.66 | 0.384 | 10.09 | 0.371 | 79.31±0.32 |
| 14 | 2960 | 591 | 11.83 | 0.409 | 11.09 | 0.3935 | 80.22±1 |
| 15 | 2868 | 637 | 13.49 | 0.44 | 12.339 | 0.419 | 80.39±1.01 |
| 16 | 2762 | 690 | 15.34 | 0.47 | 13.879 | 0.448 | 79.9±0.78 |
| 17 | 2636 | 753 | 17.37 | 0.504 | 15.87 | 0.481 | 80.73±1.5 |
| 18 | 2486 | 828 | 19.98 | 0.54 | 18.503 | 0.519 | 81.93±2.12 |
| 19 | 2368 | 887 | 24.6 | 0.59 | 20.81 | 0.549 | 83.42±1.17 |
| 20 | 2232 | 955 | 26.74 | 0.61 | 23.77 | 0.581 | 84.23±1.18 |
| 21 | 2122 | 1010 | 30.396 | 0.64 | 26.44 | 0.607 | 87.1±1.73 |
| 22[a] | 2109 | 421 | 11.48 | 0.4018 | 11.09 | 0.3935 | 79.1±0.51 |

a.   The solution slab in this system is 3 nm × 3 nm × 10 nm and the simulation box is 3 nm × 3 nm × 30 nm.
b.   There is a little difference between the concentration in the bulk region and the one of the whole system due to surface effects. The values used in the main text are the ones in the bulk region



**Figures**

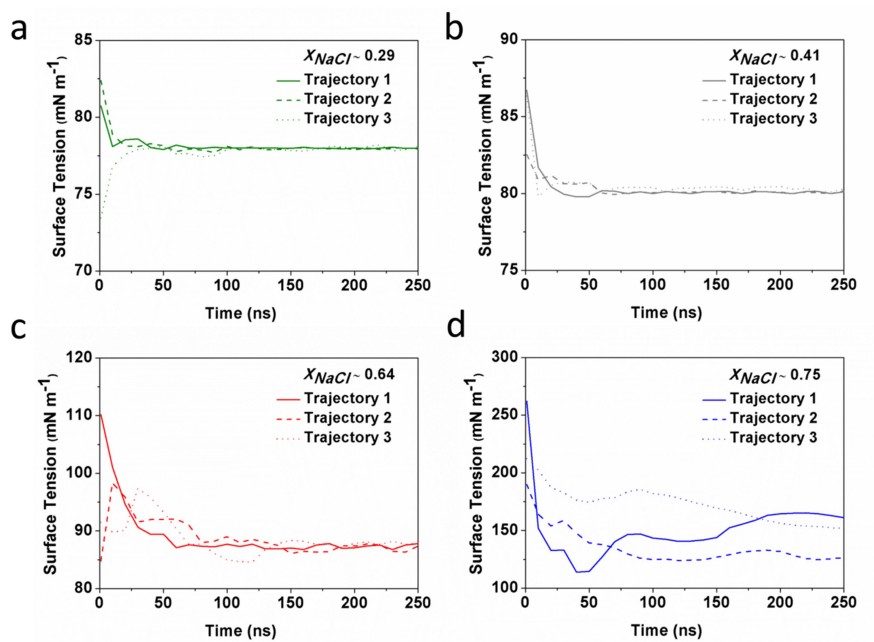

5    Figure 1. The calculated surface tension at different simulation time from different trajectories. For the solution
with $x_{NaCl} \leq 0.64$ (a-c), the surface tension become steadily stabilized after ~100-150 ns, and different individual
simulation runs converge to a similar result. When $x_{NaCl} > 0.64$ (d; here $x_{NaCl} = 0.75$), the surface tension keeps
fluctuating and the final values from different individual simulations can not be converged even after 250 ns.



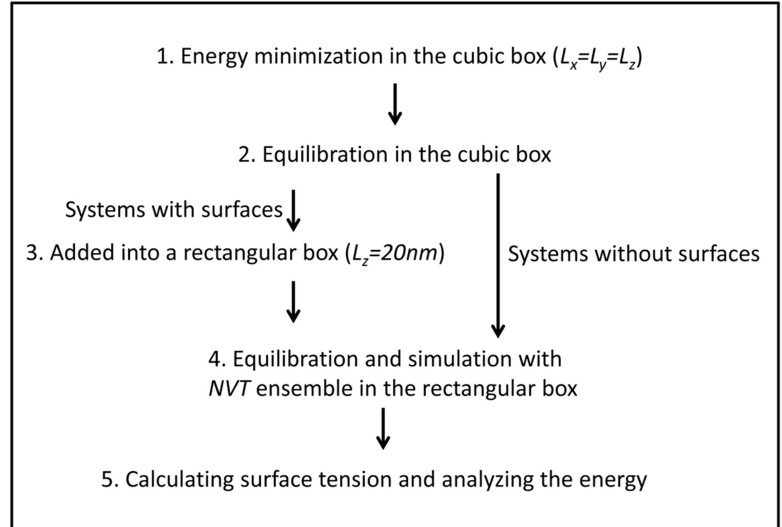

Figure 2. Schematic diagram of the different steps performed in the MD simulation.



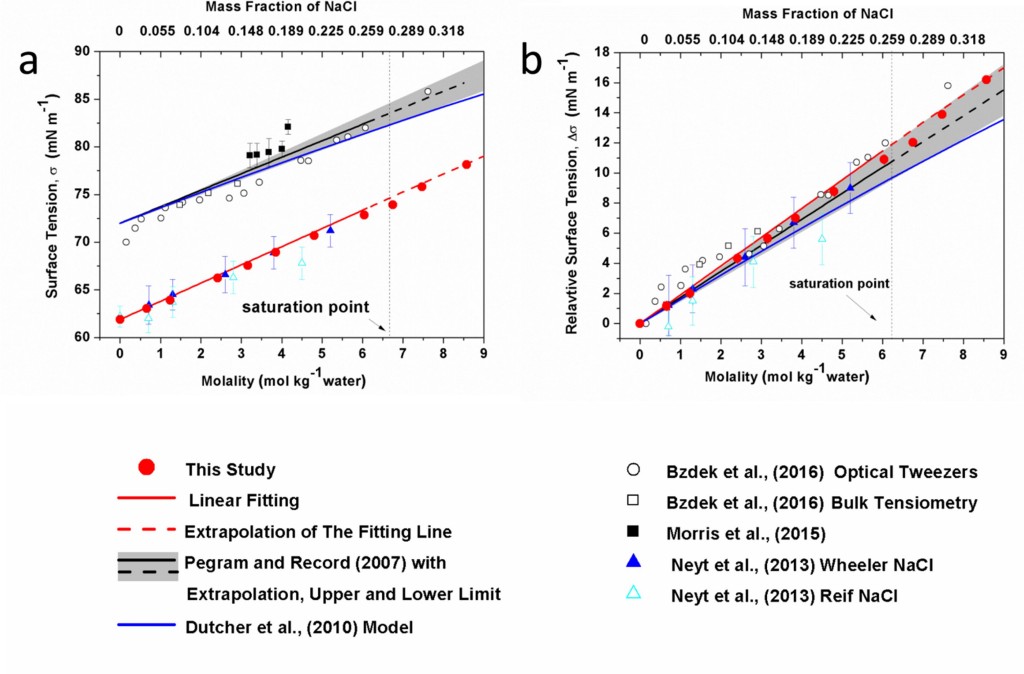

Figure 3. Surface tension (a) and relative surface tension (b) defined as $\Delta\sigma = \sigma_{solution} - \sigma_{water}$ as a function of the concentration of NaCl. The $\sigma_{water}$ in the Morris et al. (2015) study was not determined, thus the corresponding $\Delta\sigma$ is not shown in panel b.



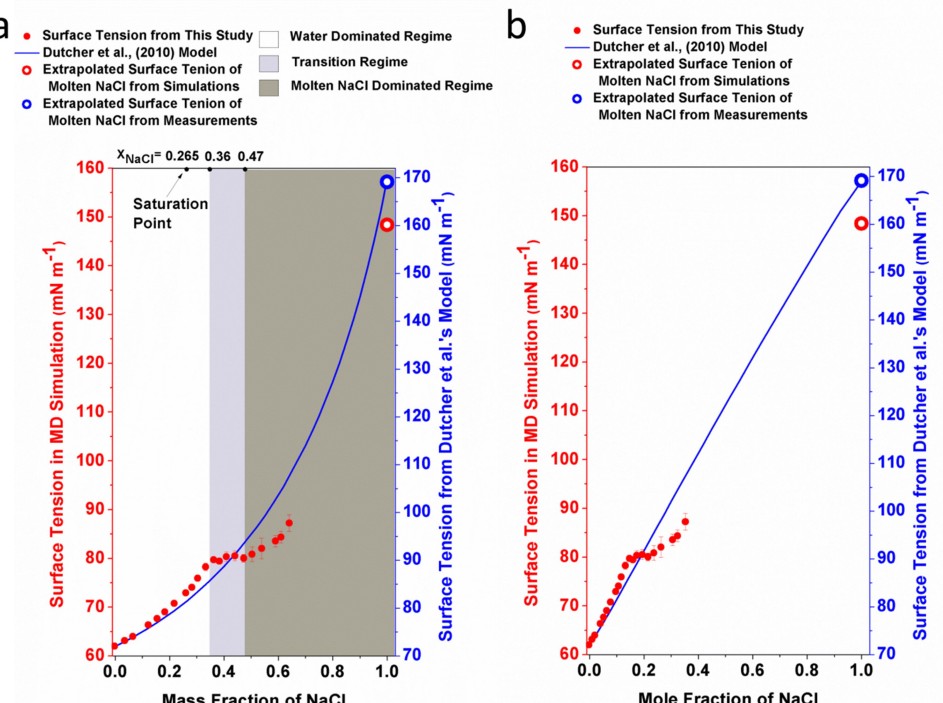

Figure 4. The surface tension of different-concentration NaCl solution. (a) The surface tension of NaCl solution
against the mass fraction of NaCl. The left red y-axis is for the data from MD simulation (red circle), and the right
blue y-axis is for the Dutcher et al. model (2010, blue solid line). The white, light grey and dark grey areas shade
the water-dominated, transition and molten NaCl-dominated regimes, respectively. (b) The surface tension of NaCl
solution is plotted against the mole fraction of NaCl.



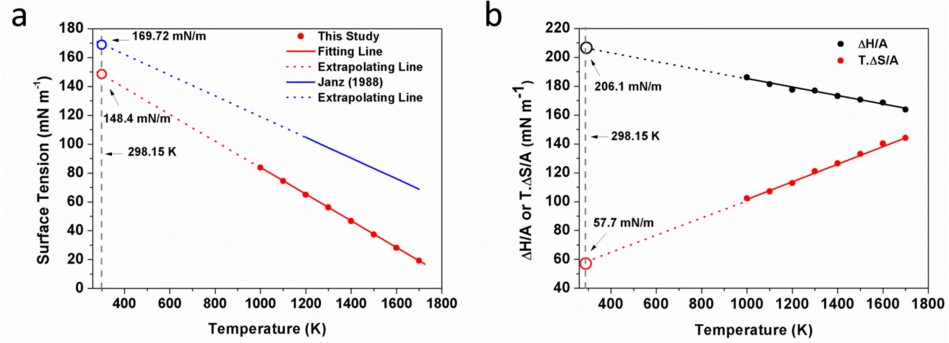

Figure 5. The surface tension, enthalpic part ($\frac{\Delta H}{A}$) and entropic part ($\frac{T \cdot \Delta S}{A}$) of molten NaCl at different temperature. (a) The equation in Janz's study (1988) is $\sigma_{NaCl} = -0.07188 \cdot T + 191$ (blue solid line). The fitting line based on our data (red solid line) is $\sigma_{NaCl} = -0.0922 \cdot T + 175.91$. The red and blue open circles represent the extrapolated value of surface tension in simulation and reality. (b) The black and red open circles represent the extrapolated $\frac{\Delta H}{A}$ and $\frac{T \cdot \Delta S}{A}$ of molten NaCl at 298.15 K, respectively.





Figure 6. The excessive surface enthalpy and entropy per unit area ($\frac{\Delta H}{A}$ and $\frac{T \cdot \Delta S}{A}$) of different NaCl solution concentrations. $\frac{\Delta H}{A}$ (black circles) and $-\frac{T \cdot \Delta S}{A}$ (red circles) are shown as a function of mass fraction of NaCl. The solid circles are obtained from simulation directly, and the open circles are obtained from the extrapolation of corresponding properties of molten NaCl. The cyan dashed line is only an auxiliary line for clearer view. Shaded areas are the same as in Figure 4.





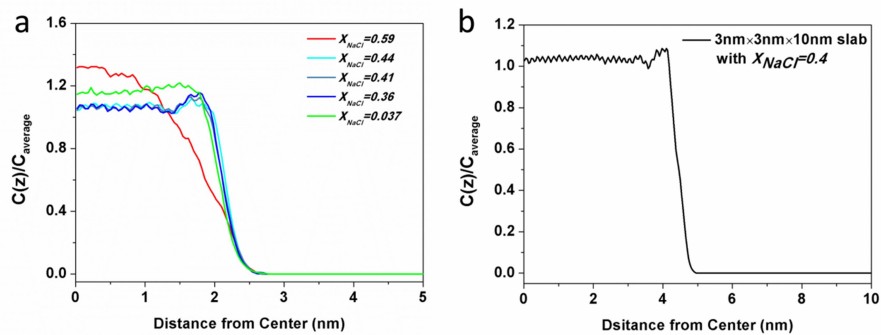

Figure 7. The ratio of Na$^+$ concentration at different positions ($C_z$) to the average concentration of the whole system ($C_{average}$). (a) The solution with $x_{NaCl}$ = 0.59 (red line) is on behalf of the solution in the molten NaCl-dominated regime (red line), the solution $x_{NaCl}$ = 0.44, 0.41 and 0.36 (blue lines) represent the solution in transition regime, and the solution $x_{NaCl}$ = 0.037 (green line) represents the solution in the water-dominated regime. (b) The density profile obtained from a 3 nm × 3 nm × 10 nm solution slab in which NaCl mass fraction is about 0.4.