# Peer review of "Molecular Dynamics Simulation of the Surface Tension of Aqueous Sodium Chloride: from Dilute to Highly Supersaturated Solutions and Molten Salt"

_Atmospheric Chemistry and Physics, 2017_

## Short Comment (SC1) · 14 Nov 2017

On p. 4, line 15, 3 references are given for the solubility of the SPC/E-compatible NaCl force field of Joung and Cheatham (JC):

The value is correctly given as $3.7 \pm 0.2$. However, the first reference (Paluch et al, 2010) provides a result (which is incorrect) for a different force field. The paper of Aragones et al (2012) gives an incorrect result for the JC force field. The correct value of $3.7 \pm 0.2$ is provided only in the final reference (Espinosa et al, 2016).

[Figure]

The first two references should be omitted, since the first is irrelevant and the second gives an incorrect result.

The history of the attempts to correctly calculate the aqueous solubility for the JC force field at 298.15K and 1 bar may be of interest.

The correct value of $3.7 \pm 0.2$ was first correctly calculated by my group:
author = Moučka, F. and Nezbeda, I. and Smith, W. R., title = Molecular Force Field Development for Aqueous Electrolytes: 1. Incorporating Appropriate Experimental Data and the Inadequacy of Simple Electrolyte Force Fields Based on Lennard–Jones and Point Charge Interactions with Lorentz–Berthelot Rules, journal = J. Chem. Theory Comput., volume = 9, number = 11, pages = 5076-5085, year = 2013

Our result was later corroborated by the Panagiotopoulos group:
author = Mester, Z. and Panagiotopoulos, A. Z., title = Mean ionic activity coefficients in aqueous NaCl solutions from molecular dynamics simulations, journal = J. Chem. Phys., volume = 142, number = 4, pages = 044507, year = 2015
and by Aragones et al. (2012) and Espinosa et al. (2016).

The history of the attempts to correctly calculate the quantity by molecular simulation are described in the following review article:
author = Nezbeda, I. and Moučka, F. and Smith, W. R., title = Recent progress in molecular simulation of aqueous electrolytes: force fields, chemical potentials and solubility, journal = Molec. Phys., volume = 114, number = 11, pages = 1665-1690, year = 2016
* * *

---

## Referee Comment (RC1) · Anonymous Referee #1 · 11 Dec 2017

This succinct paper uses molecular dynamics to probe the concentration dependent variation in surface tension of aqueous sodium chloride solutions, and to demonstrate/support the general concept of a previous semi-empirical model by Dutcher et al (2010).

I support the presentation of supporting evidence from molecular dynamics simulations in ACP generally. These tools, whilst often not overly reliable for absolute values such as saturation vapour pressures, are useful in determining trends at least. I am not an expert in MD simulations but, without providing a detailed critique of model configura-

tions, I suggest the paper is accepted for publication in ACP after some minor issues are addressed, and the paper better reshaped for the audience of this journal.

General comments:

In section 2.1, the authors note how simulations from 1000K to 1700K are used and extrapolated down to 298.15K. Is this a requirement from the simulation over simulations at lower temperatures? It is not clear whether any extrapolation would need to account for specific non-linearities that change over such a large temperature range. One might imagine any error in this process might impact on the offset presented in figure 3a?

It would be nice to see some quantitative analysis of potential impact of this work. Whilst the impact of cloud activation processes should be small, where do the authors suggest this new dependency needs to be taken into account? For example given the below cloud focus, would it potentially influence the efflorescence transition RH according to the energy differential between a solid and saturated state? Would it affect growth rates in varying humidity environments? Could you perform some quantitative analysis on this? If not, please make it clear why.

How applicable would the model be to other salts, particularly mixed salts that might arise in non-marine environments? The increased interest in bulk to surface partitioning studies require more thorough supporting studies on systems with surfactant organics. Where do MD simulations have a role here? Please guide the reader on some broad issues as to where you might demonstrate these tools in more obviously pressing issues.

Minor comments:

Page 2, line 10: I would suggest - size-effects at 'the' nanoscale.

Page 2, line 38: Suggest - based on the 'following' concept

Page 2, line 40: "solute" (t)hat

---

## Referee Comment (RC2) · Anonymous Referee #2 · 22 Feb 2018

In this paper, the authors use MD simulations to study the surface tension of NaCl aqueous solutions across all concentrations. The authors present results with an intriguing three regime behavior (water-dominated, transition, and molten salt dominated), which appears well supported by the method used. Overall, this is a good and interesting paper and I'd recommend for publication after the authors address a couple of points.

A couple of comments and suggestions:

[Figure]

Can the authors comment further on other systems such as KCl, NH4Cl, NaNO3, and NH4NO3, at least qualitatively? What about mixed-salt systems? Are the same behaviors expected?

In the transition regime, is there any reason entropy is increasing as the mass fraction approaches the efflorescence point?Âă

---

## Author Comment (AC1) · 29 Oct 2018

**Response to Comments on "Molecular Dynamics Simulation of the Surface Tension of Aqueous Sodium Chloride: from Dilute to Highly Supersaturated Solutions and Molten Salt" by Wang et al.**

Dear Editor,

Many thanks for the kind effort guiding our manuscript through the peer review process. We would also like to thank the reviewers and Dr. W. R. Smith for the valuable and constructive comments, which help us improving the manuscript. Listed below are our point-by-point responses to the comments, including the corresponding changes made to the revised manuscript. The reviewer's comments are marked in blue and our answers are marked in black, and the revision in the manuscript in further formatted as '*Italics*'.

During the manuscript revision, we discovered an error in the submitted manuscript when using the pressure tensor method to calculate surface tension (please find more details below). We sincerely apologize for it. The results have been corrected in the revised manuscript and our major finding and conclusions remain unaffected. Besides, we updated the method to determine the excess surface entropy and enthalpy of molten NaCl at 298.15 K according to the recent literature findings (Sega et al. 2018).

Thank you and best regards, Xiaoxiang Wang On behalf of all co-authors

**Technical correction**

We discovered an error in the submitted manuscript when using the pressure tensor method to calculate surface tension. Based on the diagonal component of the pressor tensors ( $P_{xx}$ ,  $P_{yy}$ ,  $P_{zz}$ ) from Molecular Dynamics (MD) simulations, the correct equation to calculate the surface tension should be Eq. R1 (Eq. 1 in the submitted and revised manuscript). However, we mistakenly applied a negative  $\langle P_{zz} \rangle$  instead of a positive one (marked in red in Eq. R2) when processing the pressor tensors data. Here  $\langle ... \rangle$  refers to the time average. For double check, all cases have been re-simulated and the results have been corrected in the revised manuscript. Since the absolute value of  $\langle P_{zz} \rangle$  is much smaller than  $\langle P_{xx} \rangle$  and  $\langle P_{yy} \rangle$  in general, the corrections to the surface tension values are relatively small (Table R1 and Figure R1) and our major finding and conclusions remain unaffected.

$$\sigma_{MD} = 0.5L_z[\langle P_{zz} \rangle - 0.5(\langle P_{xx} \rangle + \langle P_{yy} \rangle)]$$
(Eq. R1)

$$\sigma_{MD} = 0.5L_z \left[-\langle P_{zz} \rangle - 0.5 \left(\langle P_{xx} \rangle + \langle P_{yy} \rangle\right)\right]$$
(Eq. R2)

| NO. | x NaCl in
bulk region | Corrected
surface tension
(mN m -1 ) | Surface
tension in the
submitted
manuscript
(mN m -1 ) | NO.             | x NaCl in
bulk region | Corrected
surface tension
(mN m -1 ) | Surface
tension in the
submitted
manuscript
(mN m -1 ) |
|-----|-------------------------------------|-------------------------------------------------------|-------------------------------------------------------------------------------|-----------------|-------------------------------------|-------------------------------------------------------|-------------------------------------------------------------------------------|
| 1   | 0                                   | 62.24±0.044                                           | 61.9±0.02                                                                     | 12              | 0.36                                | 84.35±0.143                                           | 79.58±0.38                                                                    |
| 2   | 0.037                               | 63.48±0.03                                            | 63±0.24                                                                       | 13              | 0.384                               | 85.67±0.183                                           | 79.31±0.32                                                                    |
| 3   | 0.067                               | 64.8±0.014                                            | 63.9±0.14                                                                     | 14              | 0.409                               | 86.9±0.04                                             | 80.22±1                                                                       |
| 4   | 0.123                               | 67.41±0.089                                           | 66.23±0.1                                                                     | 15              | 0.44                                | 87.83±0.25                                            | 80.39±1.01                                                                    |
| 5   | 0.156                               | 69.49±0.006                                           | 67.56±0.17                                                                    | 16              | 0.47                                | 88.03±0.88                                            | 79.9±0.78                                                                     |
| 6   | 0.184                               | 70.76±0.1                                             | 68.93±0.06                                                                    | 17              | 0.504                               | 88.77±0.42                                            | 80.73±1.5                                                                     |
| 7   | 0.219                               | 73.61±0.055                                           | 70.67±0.1                                                                     | 18              | 0.54                                | 90.35±0.6                                             | 81.93±2.12                                                                    |
| 8   | 0.261                               | 76.06±0.14                                            | 73±0.087                                                                      | 19              | 0.59                                | 93.4±2.157                                            | 83.42±1.17                                                                    |
| 9   | 0.283                               | 77.5±0.11                                             | 73.93±0.37                                                                    | 20              | 0.61                                | 97.6±1.46                                             | 84.23±1.18                                                                    |
| 10  | 0.304                               | 79.7±0.19                                             | 75.8±0.25                                                                     | 21              | 0.64                                | 102.53±0.46                                           | 87.1±1.73                                                                     |
| 11  | 0.334                               | 82.06±0.25                                            | 78.13±0.73                                                                    | 22 a | 0.4018                              | 86.9±0.59                                             | 79.1±0.51                                                                     |

Table R1. Comparison of the corrected values of surface tension (with Eq. R1) in the revised manuscript and the ones (with Eq. R2) in the submitted manuscript.

a. The solution slab in this system is  $3 \text{ nm} \times 3 \text{nm} \times 10 \text{ nm}$  and the simulation box is  $3 \text{ nm} \times 30 \text{ nm}$ .

Figure R1. Surface tension of aqueous NaCl solution at different concentrations. (a) the corrected Figure 4a with Eq. R1 and (b) the original version with Eq. R2 in the submitted manuscript.

**Updated method to determine entropy and enthalpy of the molten NaCl at 298.15 K**

There are three ways to calculate the excess surface entropy, i.e. the direct method, the numerical derivative and the derivative of temperature-surface tension  $(T - \sigma)$  relation. Descriptions about these three methods are summarized in the Table R2. In our paper, we calculated the excess surface entropy by the direct method (1) at 298.15 K for NaCl solution up to mass fraction ( $x_{NaCl}$ ) of ~ 0.64 (Fig. 6) and (2) at high temperature of 1000 to 1700 K for molten NaCl, from which the excess surface entropy of molten NaCl at 298.15 K was extrapolated (original Fig. 5b). However, a very recent paper (Sega et al., 2018) compared these three methods in determining the excess surface entropy of liquids and found that the direct method might not be applicable at high temperature because of its significant deviations to the excess surface entropy derived with the derivative of  $T - \sigma$  relation when the temperature is high. We thus carefully checked the excess surface entropy of molten NaCl at 1000-1700 K determined from the direct method in our study. Fig. 5a shows an almost perfect linear relationship between the MD simulated surface tension of molten NaCl and temperature between 1000-1700 K  $(\sigma_{\text{molten NaCl}}(T) = -0.0755 \times T + 198.09)$ . Following Dutcher et al. (2010), we thus performed a linearly extrapolation to these data to obtain the surface tension of molten NaCl at the room temperature (298.15 K). Since  $\sigma_{\text{molten NaCl}}(T) = -0.0755 \times T + 198.09$ , by performing the derivative of T –  $\sigma$  relation ( $\frac{\Delta S(T)}{A} = \frac{-d\sigma(T)}{d^T}$ , Table R2), we can obtain an excess surface entropy ( $\frac{\Delta S_{\text{molten NaCl}}}{A}$ ) equals to 0.0755 mN m-1 K-1. This value is quite different from the slope of the data in Fig. 5b, which indicates that Sega et al. (2018)'s conclusions are also applied to our case. Therefore, we abandoned Fig. 5b in the revised manuscript. The excess entropy term  $(T \cdot \frac{\Delta S_{\text{molten NaCl}}}{A})$  of the molten NaCl at 298.15 K is directly calculated by multiplying the  $\frac{\Delta S_{\text{molten NaCl}}}{A}$  (=0.0755 mN m-1 K-1) by the temperature of 298.15 K. The entropy and enthalpy terms at NaCl mass fraction of 1.0 in Fig. 6 have thus been updated.

Note again that the majority of data in Fig. 6 (except the points for  $x_{NaCl}$  of 1.0) are obtained by the direct method at 298.15 K. We also performed independent calculation of the excess surface entropy and enthalpy of pure water at temperatures from 278.15 K to 348.15 K based on the aforementioned three methods (in Table R2). As shown in Figure R2 (Fig. S1 in the supplement of the revised manuscript), results from these three methods well agree with each other, which means that results based on the direct method at room temperature can be trusted.

Corresponding to the changes in Fig. 5 and Fig. 6, the following text was added into Page 8 Line 9-14 to introduce these calculations. "According to Fig. 5, we have  $\sigma_{NaCl} = -0.0755 \cdot T + 198.09$ , then we can get  $\frac{\Delta S_{NaCl}}{A} = 0.0755 \text{ mN m}^{-1} \text{ K}^{-1}$  because of  $\frac{\Delta S(T)}{A} = \frac{-d\sigma(T)}{dT}$  (Landau and Lifshitz, 1969). Therefore, for molten NaCl ( $x_{NaCl} = 1.0$ ),  $\frac{T \cdot \Delta S_{NaCl}}{A}$  at 298.16 K is 22.15 mN m-1, and  $\frac{\Delta H_{NaCl}}{A}$  at 298.15 K is 198.09 mN m-1 (Fig. 6). Here, we used the derivative of temperature-surface tension relation to calculate the excess surface entropy, and more discussions about the comparison of these methods can be found in the supplement (Fig. S1)".

---

## Author Response (AR1)

**Response to Comments on "Molecular Dynamics Simulation of the Surface Tension of Aqueous Sodium Chloride: from Dilute to Highly Supersaturated Solutions and Molten Salt" by Wang et al.**

Dear Editor,

Many thanks for the kind effort guiding our manuscript through the peer review process. We would also like to thank the reviewers and Dr. W. R. Smith for the valuable and constructive comments, which help us improving the manuscript. Listed below are our point-by-point responses to the comments, including the corresponding changes made to the revised manuscript. The reviewer's comments are marked in blue and our answers are marked in black, and the revision in the manuscript in further formatted as '*Italics*'.

During the manuscript revision, we discovered an error in the submitted manuscript when using the pressure tensor method to calculate surface tension (please find more details below). We sincerely apologize for it. The results have been corrected in the revised manuscript and our major finding and conclusions remain unaffected. Besides, we updated the method to determine the excess surface entropy and enthalpy of molten NaCl at 298.15 K according to the recent literature findings (Sega et al. 2018).

Thank you and best regards,
Xiaoxiang Wang
On behalf of all co-authors

**Technical correction**

We discovered an error in the submitted manuscript when using the pressure tensor method to calculate surface tension. Based on the diagonal component of the pressor tensors ($P_{xx}$, $P_{yy}$, $P_{zz}$) from Molecular Dynamics (MD) simulations, the correct equation to calculate the surface tension should be Eq. R1 (Eq. 1 in the submitted and revised manuscript). However, we mistakenly applied a negative $<P_{zz}>$ instead of a positive one (marked in red in Eq. R2) when processing the pressor tensors data. Here $<\ldots>$ refers to the time average. For double check, all cases have been re-simulated and the results have been corrected in the revised manuscript. Since the absolute value of $<P_{zz}>$ is much smaller than $<P_{xx}>$ and $<P_{yy}>$ in general, the corrections to the surface tension values are relatively small (Table R1 and Figure R1) and our major finding and conclusions remain unaffected.

$$\sigma_{MD} = 0.5L_z[\langle P_{zz} \rangle - 0.5(\langle P_{xx} \rangle + \langle P_{yy} \rangle)] \tag{Eq. R1}$$

$$\sigma_{MD} = 0.5L_z[-\langle P_{zz} \rangle - 0.5(\langle P_{xx} \rangle + \langle P_{yy} \rangle)] \tag{Eq. R2}$$

Table R1. Comparison of the corrected values of surface tension (with Eq. R1) in the revised manuscript and the ones (with Eq. R2) in the submitted manuscript.

| NO. | $x_{NaCl}$ in bulk region | Corrected surface tension (mN m$^{-1}$) | Surface tension in the submitted manuscript (mN m$^{-1}$) | NO. | $x_{NaCl}$ in bulk region | Corrected surface tension (mN m$^{-1}$) | Surface tension in the submitted manuscript (mN m$^{-1}$) |
|---|---|---|---|---|---|---|---|
| 1 | 0 | 62.24±0.044 | 61.9±0.02 | 12 | 0.36 | 84.35±0.143 | 79.58±0.38 |
| 2 | 0.037 | 63.48±0.03 | 63±0.24 | 13 | 0.384 | 85.67±0.183 | 79.31±0.32 |
| 3 | 0.067 | 64.8±0.014 | 63.9±0.14 | 14 | 0.409 | 86.9±0.04 | 80.22±1 |
| 4 | 0.123 | 67.41±0.089 | 66.23±0.1 | 15 | 0.44 | 87.83±0.25 | 80.39±1.01 |
| 5 | 0.156 | 69.49±0.006 | 67.56±0.17 | 16 | 0.47 | 88.03±0.88 | 79.9±0.78 |
| 6 | 0.184 | 70.76±0.1 | 68.93±0.06 | 17 | 0.504 | 88.77±0.42 | 80.73±1.5 |
| 7 | 0.219 | 73.61±0.055 | 70.67±0.1 | 18 | 0.54 | 90.35±0.6 | 81.93±2.12 |
| 8 | 0.261 | 76.06±0.14 | 73±0.087 | 19 | 0.59 | 93.4±2.157 | 83.42±1.17 |
| 9 | 0.283 | 77.5±0.11 | 73.93±0.37 | 20 | 0.61 | 97.6±1.46 | 84.23±1.18 |
| 10 | 0.304 | 79.7±0.19 | 75.8±0.25 | 21 | 0.64 | 102.53±0.46 | 87.1±1.73 |
| 11 | 0.334 | 82.06±0.25 | 78.13±0.73 | 22[a] | 0.4018 | 86.9±0.59 | 79.1±0.51 |

a. The solution slab in this system is 3 nm × 3nm × 10 nm and the simulation box is 3 nm × 3nm × 30 nm.

[Figure]

Figure R1. Surface tension of aqueous NaCl solution at different concentrations. (a) the corrected Figure 4a with Eq. R1 and (b) the original version with Eq. R2 in the submitted manuscript.

**Updated method to determine entropy and enthalpy of the molten NaCl at 298.15 K**

There are three ways to calculate the excess surface entropy, i.e. the direct method, the numerical derivative and the derivative of temperature-surface tension $(T - \sigma)$ relation. Descriptions about these three methods are summarized in the Table R2. In our paper, we calculated the excess surface entropy by the direct method (1) at 298.15 K for NaCl solution up to mass fraction ($x_{NaCl}$) of ~ 0.64 (Fig. 6) and (2) at high temperature of 1000 to 1700 K for molten NaCl, from which the excess surface entropy of molten NaCl at 298.15 K was extrapolated (original Fig. 5b). However, a very recent paper (Sega et al., 2018) compared these three methods in determining the excess surface entropy of liquids and found that the direct method might not be applicable at high temperature because of its significant deviations to the excess surface entropy derived with the derivative of $T - \sigma$ relation when the temperature is high. We thus carefully checked the excess surface entropy of molten NaCl at 1000-1700 K determined from the direct method in our study. Fig. 5a shows an almost perfect linear relationship between the MD simulated surface tension of molten NaCl and temperature between 1000-1700 K ($\sigma_{molten\ NaCl}(T) = -0.0755 \times T + 198.09$). Following Dutcher et al. (2010), we thus performed a linearly extrapolation to these data to obtain the surface tension of molten NaCl at the room temperature (298.15 K). Since $\sigma_{molten\ NaCl}(T) = -0.0755 \times T + 198.09$, by performing the derivative of $T - \sigma$ relation ($\frac{\Delta S(T)}{A} = \frac{-d\sigma(T)}{dT}$, Table R2), we can obtain an excess surface entropy ($\frac{\Delta S_{molten\ NaCl}}{A}$) equals to 0.0755 mN m$^{-1}$ K$^{-1}$. This value is quite different from the slope of the data in Fig. 5b, which indicates that Sega et al. (2018)'s conclusions are also applied to our case. Therefore, we abandoned Fig. 5b in the revised manuscript. The excess entropy term ($T \cdot \frac{\Delta S_{molten\ NaCl}}{A}$) of the molten NaCl at 298.15 K is directly calculated by multiplying the $\frac{\Delta S_{molten\ NaCl}}{A}$ (=0.0755 mN m$^{-1}$ K$^{-1}$) by the temperature of 298.15 K. The entropy and enthalpy terms at NaCl mass fraction of 1.0 in Fig. 6 have thus been updated.

Note again that the majority of data in Fig. 6 (except the points for $x_{NaCl}$ of 1.0) are obtained by the direct method at 298.15 K. We also performed independent calculation of the excess surface entropy and enthalpy of pure water at temperatures from 278.15 K to 348.15 K based on the aforementioned three methods (in Table R2). As shown in Figure R2 (Fig. S1 in the supplement of the revised manuscript), results from these three methods well agree with each other, which means that results based on the direct method at room temperature can be trusted.

Corresponding to the changes in Fig. 5 and Fig. 6, the following text was added into Page 8 Line 9-14 to introduce these calculations. *"According to Fig. 5, we have $\sigma_{NaCl} = -0.0755 \cdot T + 198.09$, then we can get $\frac{\Delta S_{NaCl}}{A} = 0.0755$ mN m$^{-1}$ K$^{-1}$ because of $\frac{\Delta S(T)}{A} = \frac{-d\sigma(T)}{dT}$ (Landau and Lifshitz, 1969). Therefore, for molten NaCl ($x_{NaCl} = 1.0$), $\frac{T \cdot \Delta S_{NaCl}}{A}$ at 298.16 K is 22.15 mN m$^{-1}$, and $\frac{\Delta H_{NaCl}}{A}$ at 298.15 K is 198.09 mN m$^{-1}$ (Fig. 6). Here, we used the derivative of temperature-surface tension relation to calculate the excess surface entropy, and more discussions about the comparison of these methods can be found in the supplement (Fig. S1)".*

[Figure]

Figure R2. $\frac{\Delta H}{A}$ and $\frac{T \cdot \Delta S}{A}$ of pure water at temperatures from 278.15 K to 348.15 K obtained from different methods.

Table R2. Descriptions of different methods to calculate $\frac{\Delta H}{A}$ and $\frac{T \cdot \Delta S}{A}$.

| *1. The Direct Method* |
| --- |
| We simulated liquid layers with and without surfaces. The difference of enthalpy per area of liquid with surfaces and the one of liquid without surfaces is the excess surface enthalpy ($\frac{\Delta H}{A}$). And $\frac{T \cdot \Delta S}{A}$ can be then calculated as $\frac{T \cdot \Delta S}{A} = \frac{\Delta H}{A} - \sigma$. |

| *2. The numerical derivative* |
| --- |
| We first calculated $\sigma$ of the studied liquid at different temperatures, then we used the equation $\sigma(T) = \sigma(T_0) + a \times (T - T_0) + b \times (T - T_0)^2$ to fit the data of $\sigma(T_0)$, $\sigma(T_0 - 10\,K)$ and $\sigma(T_0 + 10\,K)$ to get the fitting parameters a and b for a given $T_0$, i.e., $a(T_0)$ and $b(T_0)$, respectively. As $\frac{\Delta S(T)}{A} = \frac{-d\sigma(T)}{dT}$, we have $\frac{\Delta S}{A}(T_0) = -a(T_0)$. And we can get $\frac{\Delta S}{A}$ at different temperature one by one. For $\frac{\Delta H}{A}$, we can calculate by $\frac{\Delta H}{A} = \sigma + \frac{T \cdot \Delta S}{A}$. |

| *3. The derivative of $T - \sigma$ relation* |
| --- |
| In this method, we also need to calculate $\sigma$ of the studied liquid at different temperatures firstly, and then we can get an equation to describe the relationship between $\sigma$ and T, i.e. $\sigma(T)$. After that the excess surface entropy can be easily calculated by $\frac{\Delta S(T)}{A} = \frac{-d\sigma(T)}{dT}$. And similarly, $\frac{\Delta H}{A} = \sigma + \frac{T \cdot \Delta S}{A}$. |

**Reference:**

Dutcher, C. S., Wexler, A. S., and Clegg, S. L.: Surface tensions of inorganic multicomponent aqueous electrolyte solutions and melts, J Phys Chem A., 114, 12216-12230, 2010.

Sega, M., Horvai, G., and Jedlovszky, P.: On the calculation of the surface entropy in computer simulation. J. Mol. Liq., 262, 58-62, 2018.

**Response to Comments from anonymous referee #1**

**General comments**

1. In section 2.1, the authors note how simulations from 1000 K to 1700 K are used and extrapolated down to 298.15K. Is this a requirement from the simulation over simulations at lower temperatures? It is not clear whether any extrapolation would need to account for specific non-linearities that change over such a large temperature range. One might imagine any error in this process might impact on the offset presented in figure 3a?

**Response:**

Thanks to the reviewer for raising this important concern. A direct simulation of surface tension of molten NaCl at 298.15 K would not be possible, due to excessively large relaxation times of this system at this temperature (i.e., quick ions crystallization happens during simulation). It has been found that surface tensions of a very wide range of molten salts and their mixtures are well described by linear functions of temperatures over a temperature range of hundreds of degrees above the melting point (Horvath 1985, Janz 1988). Sada et al. (1984) also found for several molten salt hydrates that this linear relationship also applies to at least 5-10 °C below the melting point, without any discontinuity or change of slope. Thus, in the absence of simulation data of molten NaCl at very high degrees of supercooling (e.g., close or at room temperature), we follow the approach of Dutcher et al. (2010) and assume a linear relationship between surface tension of molten NaCl and temperature. With this approach, we could then retrieve the surface tension of molten NaCl at 298.15 K by extrapolating the simulated surface tension of molten NaCl in the temperature range of 1000 K to 1700 K, as shown in Fig. 5a. However, we agree with the reviewer that, in principle, non-linearity could still be possible at very high degrees of supercooling for the molten salts, which may introduce uncertainties to the offset obtained by the extrapolation. But to the best of our knowledge, no related study has been reported so far.

To clarify, we modified the related part in section 2.1 of the revised manuscript as "" following sentences "*According to Dutcher et al. (2010), surface tension of liquid/molten NaCl at 298.15 K (corresponding $x_{NaCl}$ is 1, infinite concentrated solution) can be regarded as the upper boundary of $\sigma_{NaCl,sol}$. However, a direct simulation of surface tension of molten NaCl at 298.15 K would not be possible, due to excessively large relaxation times of this system at this temperature. It has been found that surface tensions of a very wide range of molten salts can be well described by linear functions of temperature (Sada et al., 1984; Horvath, 1985; Janz 1988; Dutcher et al., 2010). We thus follow the approach of Dutcher et al. (2010) assuming a linear relationship between surface tension of molten NaCl and temperature. With this approach, we retrieve the surface tension of molten NaCl at 298.15 K by extrapolating the simulated surface tension of molten NaCl in the temperature range of 1000 K to 1700 K. Note that, in principle, non-linearity could still be possible at very high degrees of supercooling (e.g., close to or at room temperature) for the molten salts, which may introduce uncertainties to the offset obtained by the extrapolation.* (Page 3, line 29-38)"

2. It would be nice to see some quantitative analysis of potential impact of this work. Whilst the impact

of cloud activation processes should be small, where do the authors suggest this new dependency needs to be taken into account? For example given the below cloud focus, would it potentially influence the efflorescence transition RH according to the energy differential between a solid and saturated state? Would it affect growth rates in varying humidity environments? Could you perform some quantitative analysis on this? If not, please make it clear why.

**Response:**

Many thanks for the constructive comment. The reviewer is right, cloud activation processes are mainly related to the thermodynamic properties of diluted solutions. The thermodynamic properties, such as surface tension and water activity, for highly concentrated solution and for solute at molten state are essential for understanding the phase transition of nano particles (Cheng et al., 2015 and references therein). However, these data are difficult or even not possible to obtain due to technical difficulties. Although a transition regime ("plateau") around the concentration upon efflorescence (Fig. 4) was found, our simulation results in principle confirm the basic concept of the Dutcher et al. (2010) semi-empirical model. The MD simulations rather unfold a more detailed global landscape of concentration dependence of surface tension of aqueous NaCl solution, i.e., three regimes (a water-dominated regime, a transition regime and a molten NaCl-dominated regime) and their different driving forces, which may advance our understanding on the experiment-based findings that linear relationships between surface tensions of single inorganic electrolyte solutions may not valid for most highly soluble electrolytes (Dutcher et al., 2010). For example, surface tension of aqueous $HNO_3$ at ~298.15 K are linear only to mass fraction of $HNO_3$ ~0.2 (Weissenborn and Pugh, 1996); a clear plateau was found for the surface tension of aqueous ammonium sulfate (AS) at mass fraction of AS ~0.8 (the concentration upon efflorescence) (Cheng et al., 2015); and surface tension of aqueous NaCl clearly deviates from the linear function at molality of ~10 mol $kg^{-1}$ (Cheng et al., 2015), which is consistent with the starting concentration point of the "plateau" (Fig. 4). Our result may not exactly reflect the real mode of surface tension of NaCl solution along the concentration, but it does imply the concept of a non-monotonic change of surface tension.

Following the suggestion, we also tried to evaluate the impact of the "plateau" on the estimation of vapor pressure upon gas-particle equilibrium with Köhler theory that accounts for the Kelvin effect. According to the MD simulation, the surface tension of aqueous NaCl upon efflorescence ($x_{NaCl}$ of ~0.47) calculated by the E-AIM model should lower by ~5-6% (from ~ 93.2 mN/m to ~87.7 mN/m). For NaCl particles with diameter larger than 10 nm, the discrepancy in the vapor pressure estimations at $x_{NaCl}$ of ~0.47 would be less than 1%, however, for smaller nano particles, it will lead to an underestimation of vapor pressure up to ~10% (for NaCl particles with diameter of ~ 1nm).

3. How applicable would the model be to other salts, particularly mixed salts that might arise in non-marine environments? The increased interest in bulk to surface partitioning studies require more thorough supporting studies on systems with surfactant organics. Where do MD simulations have a role here? Please guide the reader on some broad issues as to where you might demonstrate these tools in more obviously pressing issues.

**Response:**

Many thanks for the constructive comment. Our simulation approach can be used to study other salts, mixed salts and surfactant organics, when appropriate parameters are available, i.e. force fields that describe the interactions between salts and water, and the interactions between different individual slats. Here, we compared the values of surface tension of many organic compounds from MD simulations based on OPLS-AA force field and measured values to show the ability of MD simulations (Caleman et al., 2011). In Figure R3, calculated values are plotted against the measured values, and all data points compactly located around the 1:1 line with slight tendency of underestimation, which suggests that MD simulations can predict the measured values reasonably well.

Although our simulation approach can be used to study other systems, we cannot conclude if the non-monotonic change of surface tension along concentration also applies to other salts or mixed system and surfactant organics. It is worth to notice that, although surface tensions of single inorganic electrolyte solutions are often assumed to be linear functions of concentration or molality over moderate concentration range, this linear relation may not be valid for most highly soluble electrolytes (Dutcher et al., 2010). For example, surface tension of aqueous $HNO_3$ at ~298.15 K is linear only to mass fraction of $HNO_3$ ~0.2 (Weissenborn and Pugh, 1996). In our previous study (Cheng et al., 2015), surface tensions of NaCl solution and ammonium sulfate solution were studied by using Differential Köhler Analysis. Anomaly was found on the surface tension-molality curve for both salts. Our result may not exactly reflect the real mode of surface tension of NaCl solution along the concentration, but it does imply the concept of a non-monotonic change of surface tension. Therefore, we think more studies are necessary to examine the concentration dependence of surface tension of other salts or mixed system and surfactant organics by using MD simulations.

To emphasize, we add the following sentences into the conclusion: "…*One must be aware that for nucleation processes in the atmosphere also other chemical compounds matter, and will require future study. Also, mixed salt solutions would be very interesting, and can in principle be studied with similar simulation methods as applied here; however, this task must be left to future work.* (Page 8, line 30-33)".

[Figure]

Figure R3. Correlation between calculated surface tension and measured values. All data used in Figure R3 is also summarized in Table R3. Data source: Caleman et al. (2011).

Table R3. Surface tension of 67 organic compounds from measurements and MD simulations. Data source: Caleman et al. (2011).

| No. | Name | Measured Values | Calculated Values | No. | Name | Measured Values | Calculated Values |
|---|---|---|---|---|---|---|---|
| 1 | methanoic acid | 37.13 | 32±0.4 | 35 | pentane-2,4-dione | 30.9 | 32.9±0.9 |
| 2 | nitromethane | 36.53 | 29.2±0.4 | 36 | methyl 2-methylprop-2-enoate | 24.24 | 29.4±0.7 |
| 3 | methanol | 22.07 | 20.1±0.4 | 37 | ethyl propanoate | 23.8 | 22.3±0.5 |
| 4 | 1,2-dibromoethane | 39.55 | 38±0.7 | 38 | diethyl carbonate | 25.92 | 25±0.8 |
| 5 | methylformate | 24.36 | 20.8±0.3 | 39 | pentan-1-ol | 25.36 | 19±1 |
| 6 | bromoethane | 23.62 | 18.2±0.6 | 40 | pentan-3-ol | 23.65 | 18.8±0.7 |
| 7 | N-methylformamide | 38.52 | 36.9±0.2 | 41 | pentane-1,5-diol | 46.32 | 46.8±5.3 |
| 8 | ethanol | 21.97 | 18.7±0.3 | 42 | nitrobenzene | 43.23 | 33.6±1 |
| 9 | methylsulfinylmethane | 42.92 | 42.4±0.9 | 43 | 2-methylpyridine | 33 | 27.6±0.2 |
| 10 | ethane-1,2-diamine | 41.12 | 32.5±0.7 | 44 | 3-methylpyridine | 35.04 | 29±0.5 |
| 11 | prop-2-enenitrile | 26.63 | 20.4±0.3 | 45 | 4-methylpyridine | 35.43 | 27.9±0.3 |
| 12 | 1,2-dibromopropane | 34.5 | 33.9±0.8 | 46 | cyclohexanone | 34.57 | 27.3±1.1 |
| 13 | methylacetate | 24.73 | 23.2±0.4 | 47 | hexan-2-one | 25.45 | 19.3±0.6 |
| 14 | 1-bromopropane | 25.26 | 19.4±0.2 | 48 | cyclohexanamine | 31.22 | 24±1.1 |
| 15 | N,N-dimethylformamide | 35.74 | 31.5±0.8 | 49 | 2-propan-2-yloxypropane | 17.27 | 13.2±0.2 |
| 16 | 1-nitropropane | 29.85 | 24±0.5 | 50 | 1-methoxy-2-(2-methoxyethoxy)ethane | 29.3 | 24.9±1.1 |
| 17 | 2-nitropropane | 29.29 | 24.9±0.8 | 51 | triethyl phosphate | 29.61 | 28.8±1.1 |
| 18 | dimethoxymethane | 18.75 | 17.1±0.3 | 52 | N-propan-2-ylpropan-2-amine | 19.14 | 15±0.3 |
| 19 | propan-2-amine | 17.36 | 13±0.3 | 53 | benzaldehyde | 38 | 32.5±0.5 |
| 20 | ethylsulfanylethane | 24.57 | 18.7±0.5 | 54 | toluene | 27.73 | 20.9±0.5 |
| 21 | butane-1-thiol | 25.22 | 19.9±0.3 | 55 | phenylmethanol | 35.97 | 30.9±1.9 |
| 22 | butane-1,4-diol | 45.47 | 45.6±3.4 | 56 | 2,4-dimethylpentan-3-one | 24.78 | 20.5±0.5 |
| 23 | 2-methylpropan-2-amine | 16.87 | 15.3±0.2 | 57 | heptan-2-one | 26.12 | 19.9±0.2 |
| 24 | furan | 22.65 | 19.6±0.4 | 58 | 1-phenylethanone | 39.04 | 33.4±1.2 |
| 25 | thiophene | 30.68 | 29.4±0.5 | 59 | methyl benzoate | 37.17 | 33.6±0.3 |
| 26 | 1H-pyrrole | 36.95 | 28.9±0.9 | 60 | methyl 2-hydroxybenzoate | 39.22 | 36.5±1.4 |
| 27 | ethenyl acetate | 22.03 | 27.4±0.6 | 61 | 1,2-dimethylbenzene | 29.76 | 22.8±0.2 |
| 28 | ethyl acetate | 23.39 | 23.2±0.6 | 62 | 1,2-dimethoxybenzene | 32.8 | 30.4±1.2 |
| 29 | thiolane | 33.82 | 26.9±0.6 | 63 | 2,4,6-trimethylpyridine | 33.3 | 26.4±0.4 |
| 30 | 1-bromobutane | 25.9 | 20.3±0.6 | 64 | quinoline | 42.59 | 33.7±0.9 |
| 31 | N,N-dimethylacetamide | 33.09 | 32.1±0.6 | 65 | (1-methylethyl)benzene | 27.69 | 21.4±0.4 |
| 32 | morpholine | 37.68 | 32.7±1.2 | 66 | 1,2,4-trimethylbenzene | 29.19 | 24±0.4 |
| 33 | pyridine | 36.56 | 29.1±0.8 | 67 | 2,6-dimethylheptan-4-one | 25.8 | 20.2±0.4 |
| 34 | cyclopentanone | 32.8 | 26.2±0.8 | | | | |

**Minor comments:**

1. Page 2, line 10: I would suggest - size-effects at 'the' nanoscale.

**Response:** Thanks. We have revised the manuscript accordingly. "…*Because of the energy barrier of*

*crystallization during dehydration and size-effects at the nanoscale…* (Page 2, line 8 in the revised manuscript)".

2. Page 2, line 38: Suggest - based on the 'following' concept

**Response:** Thanks. We have revised the manuscript accordingly. "…*This model is based on the following concept*… (Page 2, line 36 in the revised manuscript)"

3. Page 2, line 40: "solute" (t)hat

Response: Thanks. We have revised the manuscript accordingly. "…*while at very high salt concentration the water is considered as "solute" that is solvated by the ions*… (Page 2, line 38 in the revised manuscript)"

**Reference:**

Caleman, C., van Maaren, P J, Hong M., Hub, Jochen S., da Costa, Luciano T., and van der Spoel, David.: Force field benchmark of organic liquids: density, enthalpy of vaporization, heat capacities, surface tension, isothermal compressibility, volumetric expansion coefficient, and dielectric constant, J Chem Theory Comput., 8(1), 61-74, 2011.

Cheng, Y., Su, H., Koop, T., Mikhailov, E., and Pöschl, U.: Size dependence of phase transitions in aerosol nanoparticles, Nat. Commun., 6, 5923, doi:10.1038/ncomms6923, 2015.

Dutcher, C. S., Wexler, A. S., and Clegg, S. L.: Surface tensions of inorganic multicomponent aqueous electrolyte solutions and melts, J Phys Chem A., 114, 12216-12230, 2010.

Horvath, A. L.: Handbook of aqueous electrolyte solutions physical properties, estimation and correlation methods; Ellis Horwood series in physical chemistry, Ellis Horwood Limited: New York, 1985.

Janz, G. J.: Thermodynamic and transport properties for molten salts: correlation equations for critically evaluated density, surface tension, electrical conductance, and viscosity data, Amer Inst of Phys., 17, 1-39, 1988.

Sada E., Katoh S., and Damle, H G.: Surface tension of some molten salt hydrates by the pendant drop technique, J Chem Eng Data., 29(2), 117-119, 1984.

Weissenborn, P K., and Pugh, R J.: Surface tension of aqueous solutions of electrolytes: relationship with ion hydration, oxygen solubility, and bubble coalescence. J. Colloid Interface Sci., 184(2), 550-563, 1996.

**Response to Comments from anonymous referee #2**
**General comments**

1. Can the authors comment further on other systems such as KCl, NH4Cl, NaNO3, and NH4NO3, at least qualitatively? What about mixed-salt systems? Are the same behaviors expected?

**Response:**

Many thanks for the constructive comment. As in our response to general comment 3 of reviewer #1, our simulation approach can be used to study other salts (such as KCl, $NH_4Cl$, $NaNO_3$, and $NH_4NO_3$), as well as mixed salts and organics, when appropriate parameters are available, i.e. force fields those describe the interactions between salts and water, and the interactions between different individual slats. However, we cannot conclude if the non-monotonic change of surface tension along concentration also applies to other salts or mixed system. It is worth to notice that, although surface tensions of single inorganic electrolyte solutions are often to be linear functions of concentration or molality over moderate concentration range, this linear may not valid for most highly soluble electrolytes (Dutcher et al., 2010). For example, surface tension of aqueous $HNO_3$ at ~298.15 K are linear only to mass fraction of $HNO_3$ ~0.2 (Weissenborn and Pugh, 1996). In our previous study (Cheng et al., 2015), surface tension of NaCl solution and ammonium sulfate solution were studied by using Differential Köhler Analysis. Anomaly was also found on the surface tension-molality curve for both salts. Our result may not exactly reflect the real mode of surface tension of NaCl solution along the concentration, but it does imply the concept of a non-monotonic change of surface tension. Therefore, we think more studies are necessary to examine the concentration dependence of surface tension of other salts or mixed system and surfactant organics by using MD simulations.

To emphasis, we add the following sentences into the conclusion: "…*One must be aware that for nucleation processes in the atmosphere also other chemical compounds matter, and will require future study. Also, mixed salt solutions would be very interesting, and can in principle be studied with similar simulation methods as applied here; however, this task must be left to future work.* (Page 8, line 30-33)"

2. In the transition regime, is there any reason entropy is increasing as the mass fraction approaches the efflorescence point?

**Response:**

Many thanks for the constructive comment. We sincerely apologize that during the revision of the manuscript, we discovered an error in the submitted manuscript when using the pressure tensor method to calculate surface tension (please find more details in the Technical correction in the letter to the editor). Although our major finding and conclusions remain unaffected, the mis-calculation propagates the error into the energetic analyses and leads to the moderate increase of surface entropy in the transition regime when the solution concentration approaches the efflorescence point (Fig. R3b, Fig. 6 in the previously submitted manuscript). After re-simulating all cases and correcting the calculation of surface tension, we found that the surface entropy keeps almost unchanged, as shown in Fig. R3a (Fig. 6 in the revised manuscript and the related discussion has also been modified accordingly). We speculate this stability of surface entropy may be related to the surface enrichment zone of ions. Thus,

the following sentences were added: "*Tentatively, one may correlate the formation of the enrichment zone with the stability of the surface entropy in this region via the entropy of mixing. At the same time, the surface enhancement of ions may be related to the phenomenon of efflorescence.* (Page 8, line 9-11)".

[Figure]

Figure R3. The excess surface enthalpy and entropy per unit area ($\frac{\Delta H}{A}$ and $\frac{T \cdot \Delta S}{A}$) of different NaCl solution concentrations. (a) the corrected Figure 6 in the revised manuscript. (b) the original Figure 6 in the submitted manuscript. $\frac{\Delta H}{A}$ (black circles) and $-\frac{T \cdot \Delta S}{A}$ (red circles) are shown as a function of mass fraction of NaCl. The solid circles are obtained from simulation directly, and the open circles are obtained from the extrapolation of corresponding properties of molten NaCl. The cyan dashed line is only an auxiliary line for clearer view.

**Reference:**

Cheng, Y., Su, H., Koop, T., Mikhailov, E., and Pöschl, U.: Size dependence of phase transitions in aerosol nanoparticles, Nat. Commun., 6, 5923, doi:10.1038/ncomms6923, 2015.

Weissenborn, P K., and Pugh, R J.: Surface tension of aqueous solutions of electrolytes: relationship with ion hydration, oxygen solubility, and bubble coalescence. J. Colloid Interface Sci., 184(2), 550-563, 1996.

**Response to Interactive comment from W. R. Smith**

On p. 4, line 15, 3 references are given for the solubility of the SPC/E-compatible NaCl force field of Joung and Cheatham (JC): The value is correctly given as 3.7±0.2. However, the first reference (Paluch et al, 2010) provides a result (which is incorrect) for a different force field. The paper of Aragones et al (2012) gives an incorrect result for the JC force field. The correct value of 3.7 ±0.2 is provided only in the final reference (Espinosa et al, 2016). The first two references should be omitted, since the first is irrelevant and the second gives an incorrect result. The history of the attempts to correctly calculate the aqueous solubility for the JC force field at 298.15K and 1 bar may be of interest. The correct value of 3.7±0.2 was first correctly calculated by my group: author = Moučka, F. and Nezbeda, I. and Smith, W. R., title = Molecular Force Field Development for Aqueous Electrolytes: 1. Incorporating Appropriate Experimental Data and the Inadequacy of Simple Electrolyte Force Fields Based on Lennard–Jones and Point Charge Interactions with Lorentz–Berthelot Rules, journal = J. Chem. Theory Comput., volume = 9, number = 11, pages = 5076-5085, year = 2013 Our result was later corroborated by the Panagiotopoulos group: author = Mester, Z. and Panagiotopoulos, A. Z., title = Mean ionic activity coefficients in aqueous NaCl solutions from molecular dynamics simulations, journal = J. Chem. Phys., volume = 142, number = 4, pages = 044507, year = 2015 and by Aragones et al. (2012) and Espinosa et al. (2016). The history of the attempts to correctly calculate the quantity by molecular simulation are described in the following review article: author = Nezbeda, I. and Moučka, F. and Smith,W. R., title = Recent progress in molecular simulation of aqueous electrolytes: force fields, chemical potentials and solubility, journal = Molec. Phys., volume = 114, number = 11, pages = 1665-1690, year = 2016

**Response:**

We thank Dr. W. R. Smith for the interactive discussion and comments. Following the suggestion, we carefully explore the history of the attempts to correctly calculate the solubility of NaCl in water at 298.15 K (Nezbeda et al., 2016) and we agree that it is more appropriate to cite the paper by Moučka et al., (2013) here. We have modify the related statement accordingly as "…*The solubility at 298.15 K based on JC force field with SPC/E model has been determined as 3.7±0.2 mol kg$^{-1}$ (Moučka et al., 2013; Mester and Panagiotopoulos, 2015; Espinosa et al., 2016), which to our best knowledge is the value most close one to the experimental value of solubility (~6.15 mol kg$^{-1}$). Therefore, this force field is appropriate to be used to study the concentration dependence of properties. More details about the history of the attempts to correctly calculate the quantity by molecular simulation can be found in Nezbeda et al.'s review (2016).* (Page 4, line 18-23)"

**Reference:**

Aragones, J., Sanz, E., and Vega, C.: Solubility of NaCl in water by molecular simulation revisited, J. Chem. Phys., 136, 244508, 2012.

Espinosa, J., Young, J., Jiang, H., Gupta, D., Vega, C., Sanz, E., Debenedetti, P., and Panagiotopoulos, A.: On the calculation of solubilities via direct coexistence simulations: Investigation of NaCl aqueous solutions and Lennard-Jones binary mixtures, J. Chem. Phys., 145, 154111, 2016.

Paluch, A. S., Jayaraman, S., Shah, J. K., and Maginn, E. J.: A method for computing the solubility limit of solids: application to sodium chloride in water and alcohols, J. Chem. Phys., 133, 124504, 2010.

Mester Z., and Panagiotopoulos A Z.: Mean ionic activity coefficients in aqueous NaCl solutions from molecular dynamics simulations, J. Chem. Phys., 142(4), 044507, 2015.

Moučka F., Nezbeda I., and Smith W R.: Molecular force field development for aqueous electrolytes: 1. Incorporating appropriate experimental data and the inadequacy of simple electrolyte force fields based on Lennard-Jones and point charge interactions with Lorentz–Berthelot rules, J. Chem. Theory. Comput., 9(11), 5076-5085, 2013.

Nezbeda I., Moučka F., and Smith W R.: Recent progress in molecular simulation of aqueous electrolytes: Force fields, chemical potentials and solubility, Mol. Phys., 114(11), 1665-1690, 2016.